# Functional Imaging of Hypoxia: PET and MRI

**DOI:** 10.3390/cancers15133336

**Published:** 2023-06-25

**Authors:** Ryan C. Perez, DaeHee Kim, Aaron W. P. Maxwell, Juan C. Camacho

**Affiliations:** 1Florida State University College of Medicine, Tallahassee, FL 32306, USA; rcp15b@fsu.edu; 2Department of Diagnostic Imaging, The Warren Alpert Medical School, Brown University, Providence, RI 02903, USA; daehee_kim@brown.edu (D.K.); aaron_maxwell@brown.edu (A.W.P.M.); 3Department of Clinical Sciences, Florida State University College of Medicine, Tallahassee, FL 32306, USA

**Keywords:** hypoxia, magnetic resonance imaging, positron emission tomography, carcinogenesis, treatment resistance

## Abstract

**Simple Summary:**

The association between hypoxia, cancer aggressiveness and decreased therapeutic response is well-established. With a significant body of evidence suggesting that tumor hypoxia is a poor prognostic indicator, it is important to identify and quantify the presence and magnitude of hypoxia within the tumor microenvironment. In this review, we aim to summarize the major molecular pathways associated with tumor hypoxia as well as the currently available evidence regarding the use of positron emission tomography (PET) and magnetic resonance imaging (MRI) techniques for imaging hypoxia within the context of cancer. We also aim to propose future directions and discuss the challenges needed to be overcome in order to advance research in this field.

**Abstract:**

Molecular and functional imaging have critical roles in cancer care. Existing evidence suggests that noninvasive detection of hypoxia within a particular type of cancer can provide new information regarding the relationship between hypoxia, cancer aggressiveness and altered therapeutic responses. Following the identification of hypoxia inducible factor (HIF), significant progress in understanding the regulation of hypoxia-induced genes has been made. These advances have provided the ability to therapeutically target HIF and tumor-associated hypoxia. Therefore, by utilizing the molecular basis of hypoxia, hypoxia-based theranostic strategies are in the process of being developed which will further personalize care for cancer patients. The aim of this review is to provide an overview of the significance of tumor hypoxia and its relevance in cancer management as well as to lay out the role of imaging in detecting hypoxia within the context of cancer.

## 1. Introduction

Hypoxia is characterized by insufficient oxygen within a tissue to support metabolism. This phenomenon is commonly seen in carcinogenesis when a tumor outgrows its vascular supply. Often, cancer cells gradually become hypoxic and adapt by up regulating the production of proteins promoting cell selection and survival. Therefore, understanding hypoxia within a particular type of cancer can provide information regarding the relationship between hypoxia, cancer aggressiveness and altered therapeutic responses. The consequence of these changes is that patients with hypoxic tumors experience relatively poor outcomes. This has led the imaging community to develop noninvasive techniques to study and recognize hypoxia. The aim of this review is to provide an overview of the significance of tumor hypoxia and its relevance in cancer management; to understand the metabolic changes and molecular pathways that tumor hypoxia influences; to lay out the role of imaging, including MRI and PET in detecting hypoxia; and to highlight how these imaging techniques can influence the clinical management of patients.

## 2. Pathophysiology of Hypoxia in Cancer

### 2.1. Hypoxia and the Cellular Response

Oxygen homeostasis is of paramount importance under normal physiological conditions. Many pathological processes result in sub-physiologic levels of oxygen in the body (hypoxia), including cancer. Chronic or intermittent hypoxia, also referred to as cycling hypoxia, results in activation of several adaptive events. One such of these events is angiogenesis and neovascularization, which is largely mediated by hypoxia inducible factor 1 (HIF-1) and downstream effectors such as vascular endothelial growth factor (VEGF) [1,2]. Under normoxic conditions, HIF is inactivated by Von Hippel Landau protein (VHL) through proteasomal degradation [3]. In the setting of hypoxia, HIF-1 is stabilized and activates genes that promote angiogenesis, modulate metabolism, and promote cell survival and proliferation [4,5]. This is a mechanism by which cancer cells can proliferate and invade surrounding tissues [1,6]. The presence of cycling tumor hypoxia poses a challenge in the imaging of hypoxia. The pO_2_ of tumor regions can vary by as much as 20mmHg in minutes to hours or even days [7,8,9]. Understanding the dynamic tumor microenvironment as it relates to hypoxia is of importance when discussing the various imaging modalities available. 

### 2.2. Hypoxia and Acidosis

Tumor hypoxia is a critical contributor to acidosis in the surrounding extracellular matrix [1,10,11,12]. The anaerobic metabolism of glucose results in the production of lactic acid and accumulation of protons and carbon dioxide which further decreases the pH of the tumor microenvironment [13,14]. This is a consequence of oncogene activation, loss of tumor suppressor activity, and the ability of cancer cells to shift glucose metabolism from oxidative phosphorylation to glycolysis [15].These cells respond to the microenvironmental change brought on by enhanced glycolytic metabolism by upregulating key regulatory systems such as carbonic anhydrase 9 (CA-IX) which helps maintain the physiological intracellular pH that is necessary for cell function [16]. These observations have generated interest in identifying tumor regions that may be responsive to pH modifying therapies [17]. 

### 2.3. Hypoxia and Immune Function

Hypoxic conditions have also been shown to reduce activation of tumor-infiltrating lymphocytes, resulting in immunosuppression and evasion of immune detection [18]. This is the rationale behind blocking hypoxia-associated transcription factors [19]. For example, hypoxia has been shown to inhibit antigen uptake by dendritic cells as well as alter their chemokine expression, differentiation, adaptation and activation in inflamed tissues such as cancer [19]. Hypoxia can also affect the migration of CD8^+^ T cells in the tumor [19]. The combination of hypoxia and high lactic acid levels has also been shown to suppress the mTOR pathway, which further leads to impaired T cell function [20]. The mTOR pathway is critical for triggering autophagy, the removal of damaged cellular materials [21]. These mechanisms pose a great challenge in the era of immunotherapy as they contribute to increased expression of the inhibitory protein, programmed cell death-1 (PD-1), which correlates with T cell exhaustion and non-responsiveness to therapy [22]. 

### 2.4. Hypoxia and Treatment Resistance

Several studies have not only implicated hypoxia in tumor aggressiveness and metastasis, but also in resistance to chemotherapy and radiation therapy [23,24,25,26,27,28]. In general, conventional cytotoxic chemotherapeutic drugs target rapidly dividing cells, including cancer cells. However, hypoxia is known to induce a state of cancer cell quiescence and render these cells resistant to chemotherapy [23]. Additionally, many cytotoxic chemotherapy agents depend on the formation of free radicals, including active oxygen, which is limited under hypoxic conditions [23]. Radiation therapy is also partially dependent on the formation of reactive oxygen species [24]. In the absence of molecular oxygen, the cell-killing effects of ionizing radiation are dampened [24]. There now exists significant interest in developing novel therapies aimed at HIFs and harnessing the hypoxic tumor microenvironment for its potential therapeutic benefits [29,30,31,32,33]. It has also been hypothesized that hypoxia has a relevant role in resistance to locoregional therapy such as embolization. The current treatment algorithm for intermediate stage hepatocellular carcinoma (HCC) includes trans-arterial chemoembolization, which combines tumor embolization with intra-arterial delivery of doxorubicin [34]. Embolization induces hypoxia, generating hypoxia-adapted cancer cells orchestrated by HIF-1 [35,36,37], rendering them resistant to treatment [38,39,40,41,42,43]. Overall, oxygen-dependent toxicities will be reduced in hypoxic settings [42,43,44,45].

## 3. Imaging Hypoxia in Cancer

Molecular and functional imaging have critical roles in cancer care. Now that the relationship of tumor hypoxia and its relevance in cancer management has been established, understanding the relationship between noninvasive detection methods of hypoxia and particular types of cancer can provide new information regarding cancer aggressiveness and therapeutic responses. Several noninvasive clinically relevant imaging modalities can detect hypoxic areas within the tumor microenvironment. Additional advantages of imaging include the possibility of visualizing the entirety of the involved tissue. The most relevant techniques available to image hypoxia can be broadly categorized as molecular (i.e., positron imaging tomography (PET)) and functional (i.e., magnetic resonance imaging (MRI) [46,47]. 

A summary of the most widely used PET and MRI imaging techniques of tumor hypoxia can be found in Table 1.

### 3.1. Positron Emission Tomography (PET) 

#### 3.1.1. Techniques

Both PET and single-photon emission computed tomography (SPECT) utilize gamma ray photons to produce an image. SPECT employs the use of hypoxia-specific compounds joined with gamma-emitting radioisotopes, such as ^123^I and ^99m^Tc, to generate a signal from hypoxic areas of tumors [132]. SPECT imaging is easy to obtain but PET offers the highest specificity for the detection of hypoxic tissue and is generally preferred [133]. PET imaging is based on administering targeted drugs labeled with radioactive isotopes that are referred to as tracers [134]. By detecting the emitted radiation, the distribution of a given drug can be inferred [134]. Positrons are emitted by radioactive isotopes and travel a few millimeters before colliding with one of the electrons in surrounding tissue [134]. The two particles will then annihilate and emit two photons which are detected and used to trace their origin [134]. Several tracers have been developed to target hypoxic areas within tumors, each with their own set of advantages and practical uses in PET imaging. Observations that ^14^C-labeled derivatives of N-alkyl-2-nitroimidazoles may become trapped in hypoxic cells have formed the basis of the development of 2-nitroimidazoles as hypoxia imaging agents using radioactive tracers [135]. Modern derivates from this first tracer are referred as the Nitroimidazole family of compounds. 

Currently, ^18^F-fluromisonidazole (^18^F-FMISO) is the most widely used PET tracer derivate for detecting hypoxia in human studies [54]. It is not without its limitations and has a relatively slow uptake and washout kinetics which limit its hypoxia–normoxia contrast [53,54]. These limitations have led to other 18-F-based tracers being developed. The second-generation tracer ^18^F-FAZA (1-(5-fluoro-5-deoxy-α-D-arabinofuranosyl)-2-nitroimidazole)) was developed with an improved pharmacokinetic profile, resulting in increased uptake and improved hypoxia–normoxia contrast [53]. Further fluorine radiotracers from the nitroimidazole family that are currently being evaluated include ^18^F-fluoroetanidazole (^18^F-FETA), ^18^F-Fluoroerythronitromidazole (FETNIM), 1-(2-[^18^F]-fluoro-1-[hydroxymethyl]ethoxy)methyl-2-nitroimidazole (^18^F-RP-170), ^18^F-2-nitroimidazol-pentafluoropropyl acetamide (^18^F-EF5) and ^18^F-flortanidazole (^18^F-HX4) [69,136]. Copper-based agents have also been widely studied, such as the Cu complex with diacetyl-bis(N4-methylthiosemicarbazone) (ATSM) ligand [137,138,139,140,141,142,143], but are still limited as they have an unclear mechanism of hypoxia selectivity [90,91]. ^68^Ga [144], ^124/125/131^I [145,146,147], and ^99m^Tc [148] have also emerged as radiolabels of interest in the detection of tumor hypoxia through PET imaging. While these tracers have all been used to detect hypoxia directly, other measures have targeted endogenous markers that are upregulated in the setting of hypoxia, such as ^89^Zr radiolabeled carbonic anhydrase-IX (CA-IX) antibody G250-F(ab′)2 [149,150,151]

#### 3.1.2. Clinical Applications

Due to the known association between intra-tumoral hypoxia and treatment resistance, early utilization of hypoxia imaging has been focused on the identification of sub-regions of hypoxic tumors and their association with clinical outcomes. ^18^F-FMISO has successfully been used to detect hypoxia in glioma [48], breast cancer [49], head and neck tumors [50,51], and lung cancer [52], and has also been used to stratify patients for radiotherapy [53]. ^18^F-FMISO uptake is associated with tumor grade and expression of biomarkers of hypoxia including CA-IX and HIF-1α as well as angiogenesis markers including VEGF in gliomas [48]. In a study involving patients with estrogen receptor (ER) positive breast cancer, ^18^F-FMISO uptake and clinical outcomes after endocrine therapy consisting of letrozole had a significant correlation [49]. ^18^F-FMISO has also been shown to be able to be used as a prognostic indicator and to stratify patient risk for relapse of head and neck cancers [51]. A similar study also found that the combination of ^18^F-Fluorodeoxyglucose (^18^FDG) and ^18^F-FMISO PET could identify patients with a risk for recurrence of non-small cell lung cancer [52]. Presurgical tumor hypoxia volume measured on FMISO-PET was significantly correlated with post-resection disease-free survival and local recurrence in 23 patients with oral squamous cell carcinoma (SCC) [152] and 25 pancreatic adenocarcinoma patients [153]. Monitoring hypoxia during treatment has also been a subject of interest; one which may serve as a basis for dynamic treatment modulation. In 50 HNSCC patients undergoing radiotherapy, mid-treatment hypoxia on FMISO-PET was associated with local progression and worse overall survival [154].

More recently, its second-generation successor, ^18^F-FAZA, has also been found to be useful for patient stratification and response evaluation to hypoxic cytotoxins in head and neck cancers [59]. Its clinical use has also been demonstrated in hypoxic imaging of gliomas, rhabdomyosarcomas and lymphoma as well as lung, head, neck, cervical and rectal tumors [56,57,58,59,60,61,62,63,64]. In a study of 23 patients with primary head and neck squamous cell carcinoma (HNSCC) [155] and 38 advanced non-small cell lung cancer patients (NSCLC) [156], the degree of hypoxia measured by FAZA-PET was correlated with disease progression while being treated with chemoradiation. A prospective trial (DAHANCA 24, Danish Head and Neck Cancer Study) followed 38 patients with non-metastatic head and neck SCC for 7.8 years. It demonstrated that a higher pre-treatment tumor-to-muscle uptake ratio on FAZA-PET imaging resulted in a higher local recurrence rate [157]. The same group recently initiated a Phase II study (DAHANCA 33, NCT02976051), examining the feasibility and efficacy of dose escalation based on pre-treatment hypoxic imaging of FAZA-PET. This emerging evidence is providing a strong case for ^18^F-FAZA having more clinical and research use than its first-generation predecessor, ^18^F-FMISO [53]. This is largely attributable to its increased hydrophilicity, low lipophilicity, increased vascular clearance, and resultant improvement in hypoxia–normoxia contrast [53].

Clinical uses of FETNIM have been demonstrated in monitoring tumor hypoxia in esophageal cancer, head and neck tumors, and cervical cancer [65,66,67,158]. Despite FETNIM having a rapid renal clearance and low liver absorption [68], its clinical application is still questionable due to its relatively low tumor/non-tumor uptake ratio [69]. ^18^F-RP-170 has been demonstrated to accumulate in hypoxic areas of glioblastoma [70]. This is supported by high uptake of ^18^F-RP-170 in areas with a high HIF-1α index but is somewhat limited as some findings suggest that there is increased uptake in areas of high proliferative activity independent of tissue hypoxia [159]. Nonetheless, there is sufficient evidence to suggest that uptake of ^18^F-RP-170 is higher in hypoxic regions of gliomas [70] and lung cancer [71]. Improved hypoxic contrast and shorter time intervals before scanning have been observed when compared with ^18^F-FMISO [63]. ^18^F-EF5 is a unique tracer in that it has a high cell membrane permeability and plasma half-life due to its higher octanol-water partition coefficient [74]. One study in head and neck squamous cell carcinoma (HNSCC) patients concluded that the ability of ^18^F-EF5 to detect hypoxia in HNSCC was encouraging [72]. In this study, a higher tumor-to-muscle ^18^F-EF5 uptake ratio of 1.5 at 3 h after administration correlated with hypoxia. In preclinical tumor models, it was found that ^18^F-EF5 was predictive of response to fractionated radiotherapy in mice [160]. ^18^F-HX4 is a third generation nitroimidazole tracer [79] with an improved signal-to-noise ratio. There is evidence to suggest that, compared with ^18^F-FMISO and ^18^F-FAZA, ^18^F-HX4 demonstrates a higher maximum tumor-to-blood ratio with a half-life of about 3 h [63]. ^18^F-HX4 has been shown to preferentially accumulate in tissues with hypoxia as identified through high concentrations of histological markers for tumor hypoxia such as pimonidazole and CA-IX [161]. Clear correlations with ^18^FDG have been observed in non-small cell lung cancer [75], HNSCC, [76,77] esophageal, and pancreatic cancers [78].

Compared with ^18^FDG and fluorine-based hypoxic PET tracer, radiolabeled Cu-ATSM has been demonstrated to accumulate specifically in hypoxic areas of tumors [141]. Cu-ATSM also has a preferable pharmacokinetic profile and signal-to-noise ratio and is not taken up by the bladder, limiting interference [85,86,87,88,89]. Cu-ATSM has been successfully utilized as a PET tracer to map tumor hypoxia for several types of cancers. The results of a pilot study involving seventeen patients with advanced head and neck cancer suggested that (62)Cu-ATSM uptake may be a predictive indicator of tumor response to chemoradiotherapy [80]. Both FAZA PET and (62)Cu-ATSM values were correlated with poor overall survival, advanced stage, and tumor size of 47 NSCLC patients [81]. Pretreatment hypoxia on FMISO-PET of stage II/III breast cancer [162] and (60)Cu-ATSM-PET of cervical cancer [82] was shown to be associated with significantly worse disease-free survival and overall survival. Cu-ATSM has also been used to study rectal tumors, [83] and gliomas [84] as well as for the staging and detection of recurrent prostate cancer [85,86].

### 3.2. Magnetic Resonance Imaging (MRI)

#### 3.2.1. Techniques 

Unlike PET, MRI does not utilize ionizing radiation. Instead, MRI depends on a magnetic field and a radiofrequency pulse to generate images. The contrast between tissue types is determined by the magnetic properties of the tissue matter [163]. Several magnetic resonance methods have been introduced as applicable in the functional imaging of hypoxia. Dynamic contrast enhanced (DCE) MRI is one such method. DCE-MRI employs the use of several contrast agents, the most common of which are gadolinium (Gd) based [164]. DCE-MRI has demonstrated uses in the detection of hypoxia in xenograft models of cervical carcinoma, melanoma, and PDAC [92]. However, DCE-MRI is limited as it provides information on perfusion and permeability. Since perfusion is not the only factor influencing tumor oxygenation, the estimates may not be entirely indicative of tumor hypoxia [131]. In order to overcome the limitations of DCE-MRI in detecting tumor hypoxia, other techniques, such as tumor oxygenation level-dependent (TOLD) MRI and blood oxygenation-dependent (BOLD) MRI can be used to assess hypoxia in the tumor environment. 

Emerging evidence suggests that oxygen-enhanced magnetic resonance imaging (OE-MRI)—also known as tumor oxygenation level-dependent (TOLD) MRI and blood oxygenation-dependent (BOLD) MRI—can act as a more practical substitute for the imaging of hypoxia. MRI techniques have been shown to map hypoxic tumor regions at a reduced cost and increased availability when compared with PET imaging and do not require the use of radiolabeled tracers [102]. TOLD MRI techniques rely on the differing characteristics between oxygenated and deoxygenated regions to produce a signal that will reflect oxygen saturation within tissues [102]. TOLD MRI employs the use of 100% inhaled oxygen to induce arterial hyperoxia to perturb tumor concentrations of oxygen molecules in solution (O_2(s)_) which results in a heterogenous change in longitudinal relaxation rate (*R*_1_) that is directly related to tumor hypoxia [165,166]. BOLD utilizes differences in the paramagnetic differences between O_2(s)_ and deoxyhemoglobin to produce signals that reflect blood oxygen saturation [102,167,168]. Since blood oxygen saturation does not necessarily reflect tissue oxygenation, BOLD can only provide qualitative information on tumor oxygenation. A newer technique known as multiparametric quantitative BOLD (qBOLD) has been developed and has been shown to provide quantitative assessments of tumor oxygenation in mouse models with glioma [109]. This approach considers the contribution of the transverse relaxation parameter (T2), macroscopic field inhomogeneities, and blood volume fraction (BVf) to BOLD MRI estimates of oxygenation obtained in a brain tumor models [109].

MRI-Fluorine (^19^F MRI) is another MRI technique that has been explored for the use of assessing regional tumor hypoxia [169]. This modality is aided by perfluorocarbons (PFCs) which are a group of ^19^F-containing compounds derived from hydrocarbons by complete substitution of ^1^H with ^19^F [170]. After administration, PFCs are not metabolized by tissue and instead are cleared by circulation and then vaporized into the air through respiration [169]. Because of their high payload of ^19^F atoms, PFCs are widely used compounds for ^19^F MRI [171,172]. PFCs carry a large quantity of O_2_ and possess a fast gas exchange rate with surrounding tissue through free diffusion [170]. As a result, PFCs can provide a non-invasive measure of tissue oxygenation [173]. The precision of the ^19^F MRI method can reach 1–3 mmHg in hypoxic regions [114,169]. A limitation of this method is that blood-delivered PFC nanoparticles are primarily concentrated in well-vascularized tumor regions as opposed to the minimally perfused hypoxic regions, resulting in overestimation of tumor pO_2_. This has been combated by injecting PFCs into different tumor regions to achieve a comprehensive spatial measure of pO_2_ [114,174,175]. Probe toxicity must also be considered in clinical research [117].

A series of fluorinated Cu(II)ATSM derivatives for potential use as ^19^F magnetic resonance agents for sensing cellular hypoxia has also been introduced [176]. The synthesized complexes feature a hypoxia-targeting Cu^2+^ coordination core, and nine equivalent fluorine atoms connected via a variable-length polyethylene glycol linker. The ethylene glycol linker can effectively modulate the lipophilicity and redox properties of the complexes, leading to different cell uptake levels and selectivity between live cells grown under normoxic and hypoxic conditions [176]. Magnetic resonance spectroscopy (MRS) utilizes the quantum spin properties of ^1^H in a magnetic field to absorb and emit radiofrequency. This technique can obtain a spectrum of the concentration of metabolites resonating at different frequencies in high resolution when placed in a magnetic field [177]. Though it does not assess for tissue oxygenation directly, it can provide a measurement of metabolic products that are a result of hypoxia, such as lactate, which may also reflect treatment-induced changes in tumor oxygenation [178,179].

Electron paramagnetic resonance imaging (EPRI), also known as electron spin resonance (ESR), is another magnetic resonance modality that can be used to image hypoxia. While MRI maps the distribution of protons, EPRI/ESR measures unpaired electron spins of diffusible O_2_ using an injected spin probe to measure relaxation directly [180]. The energy released when the two unpaired electrons of molecular O_2_ collide with the probe’s unpaired electron is linearly proportional to the O_2_ concentration being measured [181]. 

EPRI/ESR has been more recently enhanced with the development of Overhauser-enhanced MRI (OMRI) which combines MRI and EPRI/ESR by using a low-field MR scanner and a paramagnetic contrast agent [130]. OMRI utilizes the Overhauser enhancement in tissue water protons that is generated when a paramagnetic agent is hyperpolarized through electromagnetic irradiation [131]. This causes a transfer of electron polarization to occur toward the surrounding water’s protons [131]. This technique has allowed for a higher resolution. Though this double imaging technique has the potential to be a powerful tool in the imaging of hypoxia, it needs to be refined before it can be implemented in a clinical setting. The major limitations of the clinical uses of OMRI are the undesired heating of the sample due to the saturation pulse and the limited amount of the needed equipment [131].

#### 3.2.2. Clinical Applications

The clinical applications of TOLD MRI are currently being investigated but this modality has been shown to be able to distinguish between radiation necrosis and residual tumor in a mouse model of malignant glioma, which reflects one potential clinical application [182]. Furthermore, potential use in radiotherapy prognosis has been suggested in a small study of rats with Dunning R3327-AT1 tumors treated with radiotherapy [183]. This study demonstrated a slower rate of tumor growth in tumors with greater oxygen content [183]. However, comparable results were not reproducible in mice bearing glioma and rhabdomyosarcoma xenografts where carbogen-induced challenges were performed, emphasizing the need for additional data [184]. A pre-clinical study using Calu6 and U87 xenografts has demonstrated that the OE-MRI biomarker ‘perfused Oxy-R’ is sensitive to changes in hypoxia induced by hypoxia-modifying targeted therapies [185]. In this study, both the hypoxia-activated cytotoxic prodrug banoxantrone and the oxygen consumption modifier atovaquone demonstrated activity in the xenograft models with reduction in the volume of tumor identified by OE-MRI when compared with control [185]. 

OE-MRI has also been utilized to identify hypoxic changes between fractionated radiotherapy in many different histologic types of tumors, including brain metastasis [186], HNSCCs [187], NSCLC [188], renal cell carcinoma (RCC) [189], glioblastoma [190] and hepatocellular carcinoma [191]. This has led to the development of a hybrid MRI-radiotherapy linear accelerator (MR Linac) system, which was validated in human subjects [97]. Hypoxia monitoring during treatment has expanded outside the field of radiation oncology and is now studied in immunotherapy [192], anti-angiogenic therapies [167,193], and hypoxia-activated prodrugs [194,195,196], as well as minimally invasive local treatments by interventional radiology [197,198]. 

There is also evidence suggesting the utility of BOLD for monitoring tumor hypoxia and predicting response to therapy in rodent models [199]. This has been demonstrated in patients with head and neck tumors and prostate cancer [103,104]. BOLD response to breathing oxygen before chemotherapy has also been demonstrated to be significantly different in tumors with good therapeutic outcomes compared with those with poor outcomes in patients with breast cancer [105]. Traditional dynamic contrast-enhanced (DCE) MRI failed to identify those differences [105]. Another study also demonstrated that anti-angiogenic agents combined with hypoxia-activated prodrugs (HAPs, Figure 1) can inhibit tumor growth effectively and that BOLD-MRI can be used to monitor tumor perfusion, hypoxia, cell apoptosis, and proliferation in colon cancer xenograft models in mice [167]. Despite these data, there are significant concerns with the use of BOLD to assess tumor hypoxia. The BOLD effect does not correlate well with absolute PO_2_ levels, is strongly dependent on perfusion, is susceptible to motion artifacts, and absolute value is influenced by adequacy of oxygen saturation during inhalation [178,200]. Perfluoropolyether (PFPE)-based hyperbranched (HBPFPE) nanoparticles with attached peptide aptamer as targeting ligands have been developed and demonstrated use in the detection of breast cancer with high ^19^F MRI [110]. 

A phase II assessment of tumor hypoxia in newly diagnosed glioblastoma patients revealed that several MRS markers predicted overall survival at 1 year and 6-month progression free survival [118]. Though the clinical utility of MRS seems promising, research is currently limited by its time-consuming nature. EPRI/ESR has shown promise in the preclinical research phase and has demonstrated a sub-millimeter resolution of PO_2_ in tissues [201]. EPRI/ESR has been used to measure tumor oxygenation after treatment with evofosfamide in pancreatic adenocarcinoma xenograft models [124]. It has also successfully assessed hypoxia in glioblastoma [125] and colon adenocarcinoma in rat models [126]. EPRI/ESR probes may last in the site of interest for several months after injection and tolerate serial imaging over several hours, which may make it useful for mapping hypoxic regions in a live animal over time [124,202]. EPRI/ESR has also been used to distinguish cycling hypoxia and chronic hypoxia in tumor-bearing mice [203]. EPRI/ESR has made it possible to obtain 3D pO_2_ maps within 3 min, enabling non-invasive imaging of cycling hypoxia in tumors [9].

### 3.3. Additional Techniques

Photoacoustic (PA) imaging is a technique that utilizes detection of light-absorbing molecules inside tissues [204,205]. PA imaging can exploit the differences in absorption spectra between oxygenated and deoxygenated hemoglobin providing estimates of hypoxia [205]. Though this technique allows for the measurement of hypoxia using endogenous contrast, oxygen sensitive dyes have also been utilized to observe hypoxic gradients within tumors using PA imaging. Hypoxic tumor gradients have been observed in prostate cancer tumors between 5–10 mm with the use of methylene blue dye and have been confirmed with a needle-mounted oxygen probe [206]. Higher resolution PA techniques have subsequently been developed, such as ratiometric PA. A ratiometric probe with an N-Oxide functionality that undergoes selective bioreduction under hypoxic conditions has been shown to measure high resolution oxygen gradients between tumor cells at centimeter depths in rats [207]. Gold nanorods (AuNRs) that contain nitroimidazole units have also been employed in the detection of tumor hypoxia in xenograft models [208]. PA imaging is limited in its ability to detect tumor hypoxia in deep tissues, though it has been used in breast cancer which may reflect a clinical application in soft tissue tumors [209].

Additional optical techniques have been used to measure hypoxia. Diffuse optical spectroscopic imaging (DOSI) has emerged as a promising optical technique for functional imaging of hypoxia through the clinical course of breast cancer treatment. DOSI uses non-ionizing near-infrared light to provide non-invasive measures of concentrations of oxyhemoglobin [210]. DOSI has demonstrated efficacy in the detection of breast cancer [211] and has been used to successfully predict neoadjuvant chemotherapy (NAC) response in patients with breast cancer [210]. Diffuse reflectance spectroscopy (DRS) is a similar technique based on light tissue interactions that can distinguish healthy tissue from malignant tissue in the operating room [212]. Spectral characteristics between invasive carcinoma, ductal carcinoma in situ and healthy tissues have been shown to be discernable with DRS, reflecting a clinical utility in determining surgical margins during breast cancer resection [212]. 

Luminescence quenching is another optical technique that uses luminescent probes to report absolute oxygen concentration by transferring energy to nearby molecular oxygen [213]. The oxygen-quenching effect is quickly reversible which makes it possible to monitor the dynamics of oxygen changes in tumors and distinguish chronically hypoxic regions from acutely hypoxic regions, which may reflect a utility in the detection of cycling hypoxia [213]. Cherenkov luminescence imaging is a new method of molecular imaging that captures visible light emission during the radioactive decay of positron-emitting radionuclides. With the assistance of probes for oxygen or hypoxia-activated molecules, Cherenkov luminescence imaging may become a useful technique in the functional imaging of hypoxia [214] and in the monitoring of tumor response to hypoxia-targeted drugs [215].

### 3.4. Invasive Techniques

The invasive polarographic electrode has been referred to as the “gold standard” for measuring tumor hypoxia [216]. Though invasive, it is not considered a high risk modality [217]. This technique involves inserting an electrode into a tumor or metastatic lymph node and measuring oxygen from several points to provide a direct measurement of tumor oxygenation. Though this measurement is direct, measurements are only taken in a point-based manner and provide very limited spatial resolution when compared with non-invasive imaging methods. In addition to the invasive nature and point-based restraints of polarographic electrodes, several other limitations have driven research into different methods that can serve as viable modalities for the measurement of tumor hypoxia. This technique is not capable of discriminating viable from necrotic tissue, and overestimates hypoxia when necrotic areas are sampled [217]. Polarographic electrodes also function poorly when patients are administered halogenated anesthetics, which influence oxygen measurements [218].

### 3.5. Hypoxia Imaging and Interventional Radiology

Few studies have incorporated the imaging of hypoxia in the context of interventional radiology. A pilot study on FMISO-PET for HCCs undergoing transcatheter arterial embolization (TAE) was undertaken in order to identify treatment-related hypoxic response [197]. This demonstrated a quantitative correlation between hypoxia and treatment response, though the clinical utility of this was questioned due to minimal measurable differences attributed to the low signal-to-noise ratio of ^18^F-FMISO in the liver [197]. Preclinical studies of the VX2 rabbit model hepatocellular carcinoma undergoing Yttrium-90 glass microsphere radioembolization revealed the potential predictive role of pre-treatment BOLD-MRI imaging for treatment outcomes [198]. Lastly, potential effects of combination therapy involving TAE and hypoxia-activated prodrugs to enhance the efficacy of embolotherapy are also under investigation [196].

## 4. Future Directions

While significant advances have been made in the applications of imaging to understand hypoxia and its role in cancer, several areas still need to be explored as well as their clinical applications. Advances in immune checkpoint inhibitors as well as non-responders to therapy highlight the importance of imaging programmed cell death-ligand 1 (PD-L1)/programmed cell death protein 1 (PD-1) within the microenvironment of any given tumor to truly understand the role of hypoxia. This interaction is being explored from the therapeutic perspective in the locoregional therapy arena [196]. However, from the imaging standpoint, a clinical study still has not been designed to our knowledge. Significant challenges in the clinical setting include the reproducibility of techniques and the resolution of currently available hypoxia imaging modalities. Defining hypoxic areas, cycling tumor hypoxia, and heterogeneity within and between tumors of the same patient are factors that significantly affect interpretation. Taking this into account, theranostics is the natural next step in the evolution of the developed radiotracers. One example of hypoxia-derived theranostics is radiation therapy dose painting. This is where a boost of radiation is delivered to hypoxic tumor regions and identified on imaging. Additionally, there is now a trend to develop novel nanoparticles or targeted small molecules with α or β-emitters for hypoxia-based radiotheranostics [219]. One difficulty is the delivery of intravenous radiotheranostics to target tumor cells. Maximizing tumor uptake and minimizing off-target toxicity has proven to be a challenge. This may expand the role of the interventional radiologist as concentration of these particles and precise delivery can improve not only their imaging capability but also their therapeutic efficacy [220].

## 5. Conclusions

Identifying hypoxia within tissues and within the tumor microenvironment non-invasively may be of great value to improve treatment and can help in understanding those cases that are non-responders or refractory to any given therapy. Multiple imaging modalities are now clinically available and under investigation. PET tracers have been investigated and developed and it is only natural that these will transition into the theranostics arena. Nevertheless, determination of the clinical utility of these tracers and techniques is still ongoing and further evidence-based data should be constructed especially in the locoregional therapy arena. Future trials should focus on the use of hypoxia imaging techniques as a prognostic biomarker. In addition, a strategy to either change therapeutic approaches earlier or to potentially target hypoxia to decrease cancer aggressiveness and course must be studied. Combinations of hypoxia-driven therapeutic strategies with immunotherapy should be explored as well, especially with the aid of locoregional therapy through interventional radiology approaches.

## Figures and Tables

**Figure 1 cancers-15-03336-f001:**
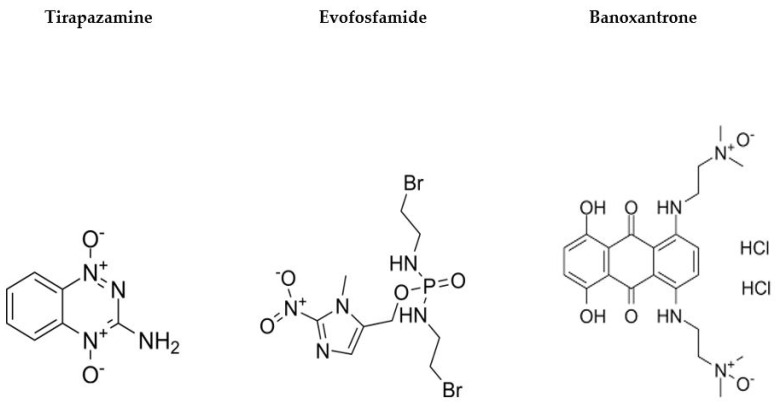
Example hypoxia-activated prodrugs (HAPs) and chemical structures.

**Table 1 cancers-15-03336-t001:** Summary of the most widely used PET and MRI imaging techniques of tumor hypoxia.

Imaging	Cancers Studied	Advantages	Disadvantages
*PET Tracers*			
PET—^18^F-FMISO	Glioma [48] Breast [49] Head and neck [50,51] Lung [52]	Most commonly used PET tracer for clinical and research applications [53]	Slow uptake and washout kinetics [53,54] Low uptake [55] 5–7 mm resolution [54]
PET—^18^F-FAZA	Glioma [56] Lymphoma [56] Lung [57,58] Head and neck [59,60] Cervical [61,62] Rhabdomyosarcoma [63] Rectal [64]	Favorable vascular clearance and improved hypoxia–normoxia contrast when compared with ^18^F-FMISO [53]	
PET—FETNIM	Esophageal [65] Head and neck [66] Cervical [67]	Rapid renal clearance and low liver absorption [68]	Low tumor/non-tumor uptake ratio [69]
PET—^18^F-RP-170	Glioma [70] Lung [71]	Favorable time interval before scanning and hypoxia contrast [71]	
PET—^18^F-EF5	Head and neck [72] Cervical [73]	High plasma half life [74]	
PET—^18^F-HX4	Lung [75] Head and neck [76,77] Esophageal [78] Pancreatic [78] Rhabdomyosarcoma [79]	High maximum tumor-to-blood ratio [63] 3-h half-life [63]	
PET—Cu-ATSM	Head and neck [80] Lung [81] Cervical [82] Rectal [83] Glioma [84] Prostate [85,86]	Favorable pharmacokinetic profile, signal-to-noise ratio, and is not taken up by the bladder [85,86,87,88,89]	Unclear mechanism of hypoxia selectivity [90,91]
** *Magnetic Resonance Techniques* **			
DCE MRI—most commonly used with gadolinium-based contrast agents	Cervical [92] Melanoma Pancreatic Breast [93] Head and neck [94]	Over-time assessment [95]	Dependent on perfusion [92] Limited resolution [96]
TOLD MRI—hyperoxic inhalation	Head and neck [97] Rectum [98] Cervix [99] Lung [100] Glioma [101]	Maps oxygen delivery in tissues [102]	Motion artifact susceptibility [95]
BOLD MRI—optional hyperoxic inhalation	Head and neck [103] Prostate [104] Breast [105] Cervical [106] Astrocytoma [107] Osteosarcoma [108]	qBOLD quantitative O_2_ mapping [109] High resolution [95]	Dependent on perfusion [109]
MRI—Fluorine—^19^F probes	Breast [110] Colon [111] Glioma [112,113] Prostate [114] Lung [115]	Quantitative PO_2_ measurement [116]	Probe toxicity [117] Low availability
MRS—endogenous lactate	Glioma [118] Breast [119,120] Head and neck [121] Prostate [122] Lung [123]	Provides quantitative measurement of metabolic byproducts of hypoxia	Time consuming
EPRI/ESR—paramagnetic probe	Pancreatic [124] Glioma [125] Colon [126]	Measuring cycling hypoxia [9]	Limited sensitivity compared with OMRI
OMRI—hyperpolarized paramagnetic contrast	Squamous cell Carcinoma [127] Colorectal [128] Breast [129]	High image resolution [130] Rapid image acquisition [130]	Limited equipment Undesired heating of sample [131]

## Data Availability

Not applicable.

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
