# Peer review of "Functional Imaging of Hypoxia: PET and MRI"

_cancers, 2023, doi:10.3390/cancers15133336_

Round 1

Reviewer 1 Report

This paper is very well written and summarizes a very relevant topic. 

Pathophysiology of Hypoxia in Cancer paragraph is currently 1 ½ page and tangent to the topic of the paper. Suggest shortening that paragraph to ½ or ¾ of page at most.  

I would add a paragraph summarizing invasive and noninvasive imaging. Optical sensing is used in the OR…It has shortcomings that limit its use to OR or superficial indications. A discussion about diffuse optical spectroscopy imaging in breast cancer would be appreciated. 

The table formatting is unclear, and it is not easy to follow the rows. 

Line 164: Please do not start a paragraph with the word “Because” 

Page 6 line 215-216: what does encouraging mean? I would specify results with objective and quantitative measures 

It would be appreciated if the authors can discuss the papers that have created controversy and knocked back acceptance of certain hypoxia imaging techniques i.e. BOLD imaging or at least explain the pitfalls and/or shortcomings

Also the authors should discuss the role of cycling tumor hypoxia and its effects on imaging 

This is a well written paper. one minor edit re: starting a paragraph with Because. 

Author Response

  1. Pathophysiology of Hypoxia in Cancer paragraph is currently 1 ½ page and tangent to the topic of the paper. Suggest shortening that paragraph to ½ or ¾ of page at most.  

The language in the “Pathophysiology of Hypoxia in Cancer” sections have been edited to be as concise as possible and will be reflected on the tracked changes.

  1. I would add a paragraph summarizing invasive and noninvasive imaging. Optical sensing is used in the OR…It has shortcomings that limit its use to OR or superficial indications. A discussion about diffuse optical spectroscopy imaging in breast cancer would be appreciated. 

A paragraph titled “Invasive Techniques” has been added to the manuscript (Line 454) to briefly review key techniques. This was not discussed further as this is an invited manuscript focused on highlighting PET and MRI.

  1. The table formatting is unclear, and it is not easy to follow the rows. 

Table 1 has been revised with a new, easier to follow format, along with additional content highlighting key advantages and disadvantages.

  1. Line 164: Please do not start a paragraph with the word “Because” 

This has been corrected and the revised sentence can be found on Line 189.

  1. Page 6 line 215-216: what does “encouraging” mean? I would specify results with objective and quantitative measures 

A sentence was added on line 241 to elaborate on the results of the cited study.

  1. It would be appreciated if the authors can discuss the papers that have created controversy and knocked back acceptance of certain hypoxia imaging techniques i.e. BOLD imaging or at least explain the pitfalls and/or shortcomings

The major shortcomings of BOLD imaging have been elaborated beginning on line 295. Additional techniques that have been produced to address these pitfalls are also discussed beginning here.

  1. Also the authors should discuss the role of cycling tumor hypoxia and its effects on imaging

An introduction to cycling tumor hypoxia was added beginning on line 65 and the imaging techniques best suited for measuring it are discussed beginning on line 405.

Thank you again for your comments and constructive feedback. 

Reviewer 2 Report

The manuscript “Functional Imaging of Hypoxia: PET and MRI” discusses current strategies for imaging of tumour hypoxia using PET and MRI techniques. First, the pathophysiology of hypoxia in the tumour microenvironment is discussed. Second, PET and MRI strategies for hypoxia imaging are introduced and their clinical applications are reviewed. Finally, future directions for advancing research in the field are highlighted.

The review needs to address several major changes: 

1.     It is mentioned in the introduction that the “role of imaging including MRI and PET” (l.45/46) is discussed in the review, but there is no comparison of PET and MRI to any other hypoxia imaging modality (such as Photoacoustic imaging, electron paramagnetic resonance imaging (EPRI), SPECT etc). A comparative table or similar should be added.

2.     A short introduction (3-4 sentences) into the basic working principles of MRI and PET is lacking.

3.     The references for the table are missing. Additionally, key techniques are missing, such as DCE-MRI (see review: doi: 10.3390/cancers12071979), MRI- fluorine, redox activated MRI hypoxia probes, etc

4.     The table would be more helpful if advantages and disadvantages (or potential/limitations) and probe techniques/mechanisms were discussed instead of featuring a “comments” column and an incomplete section of use cases.

5.     Key limitations of the imaging approaches should be more clearly outlined in the discussion section (e.g., relating to technical limitations (e.g. sensitivity/specificity), standardisation, costs, translational barriers etc.).

6.     There are fundamental differences in the oxygen measurements of the presented techniques, e.g., some measure tissue oxygenation (e.g., PET tracer), whilst others assess oxygenation in blood vessels (e.g., BOLD-MRI). This (and its implications) should be more clearly highlighted. It may be also of interest to discuss MRI and PET approaches of imaging different types of hypoxia (e.g., acute vs intermittent hypoxia).

7.     The writing needs to be improved in terms of clarity, concision, and overall quality.

8.     The general novelty of the review is unclear. There are already recent reviews on PET and MRI hypoxia imaging out there which appear much more complete and informative, e.g.,:

a.     “Advances in PET and MRI imaging of tumor hypoxia” (2023) doi: 10.3389/fmed.2023.1055062

b.     “Imaging of Tumor Hypoxia with Radionuclide-Labeled Tracers for PET” (2021), doi.org/10.3389/fonc.2021.731503

c.     “Imaging of hypoxia using PET and MRI” (2015), doi: 10.2174/138920112799436267

see above

Author Response

1. It is mentioned in the introduction that the “role of imaging including MRI and PET” (l.45/46) is discussed in the review, but there is no comparison of PET and MRI to any other hypoxia imaging modality (such as Photoacoustic imaging, electron paramagnetic resonance imaging (EPRI), SPECT etc). A comparative table or similar should be added.

SPECT has been introduced to the paper beginning on Line 149. DCE-MRI, q BOLD, MRI-Fluorine, MRS, EPRI and OMRI have been added to the “Techniques” subsection of the MRI section (Line 267). Photoacoustic imaging and optical techniques have been added under a new section (Line 416) and invasive techniques have been mentioned under a new section (Line 454). The PET and MRI sections of Table 1 have been revised to include relevant techniques as well as their key advantages and disadvantages.

  1. A short introduction (3-4 sentences) into the basic working principles of MRI and PET is lacking.

A short introduction to the working principles for PET and MRI have been added and can be found beginning on lines 153 and 263, respectively.

  1. The references for the table are missing. Additionally, key techniques are missing, such as DCE-MRI (see review: doi: 10.3390/cancers12071979), MRI- fluorine, redox activated MRI hypoxia probes, etc

References for the table and additional techniques including DCE-MRI, MRI- Fluorine, MRS, EPRI and OMRI have been added to Table 1 and under the “Techniques” subsections of PET and MRI.

  1. The table would be more helpful if advantages and disadvantages (or potential/limitations) and probe techniques/mechanisms were discussed instead of featuring a “comments” column and an incomplete section of use cases.

Table 1 has been reformatted to include imaging modalities and tracers, cancers studied along with additional information on advantages and disadvantages of each modality.

  1. Key limitations of the imaging approaches should be more clearly outlined in the discussion section (e.g., relating to technical limitations (e.g. sensitivity/specificity), standardisation, costs, translational barriers etc.).

Key advantages and disadvantages have been included in the discussion of each presented technique and are summarized in Table 1.

  1. There are fundamental differences in the oxygen measurements of the presented techniques, e.g., some measure tissue oxygenation (e.g., PET tracer), whilst others assess oxygenation in blood vessels

(e.g., BOLD-MRI). This (and its implications) should be more clearly highlighted. It may be also of interest to discuss MRI and PET approaches of imaging different types of hypoxia (e.g., acute vs intermittent hypoxia).

The differences in methods of oxygen measurement have been added to the discussion of each presented technique and are also reflected on Table 1. The advantages of EPRI on the imaging of cycling hypoxia is also now discussed beginning on line 405.

  1. The writing needs to be improved in terms of clarity, concision, and overall quality.

The manuscript has been reviewed with extensive edits regarding content and concision. We believe you will find the quality of writing to be improved throughout.

  1. The general novelty of the review is unclear. There are already recent reviews on PET and MRI hypoxia imaging out there which appear much more complete and informative, e.g.,:

a. “Advances in PET and MRI imaging of tumor hypoxia” (2023) doi: 10.3389/fmed.2023.1055062

b. “Imaging of Tumor Hypoxia with Radionuclide-Labeled Tracers for PET” (2021),

doi.org/10.3389/fonc.2021.731503

c. “Imaging of hypoxia using PET and MRI” (2015), doi: 10.2174/138920112799436267

This is an invited manuscript for a special edition on imaging of hypoxia and we are fulfilling the editorial need as per the editor’s guidance. The major contributions are the proposed applications to the field of interventional radiology which is uniquely positioned to investigate therapeutic options.

Thank you again for your comments and constructive feedback.

Reviewer 3 Report

Overall: Well written manuscript.

Interesting review article regarding functional imaging of hypoxia.

This review article represents up to date information.  Based on my PubMed and Google scholar search, it represents recent, up to date information.

Title: Appropriate

Abstract: No changes

Introduction: No changes

Discussion: 

Page 3 line 200 change "diving" cells to "dividing" cells

Please use convention of punctuation in which comma is used before the word "and" in a list of words or independent phrases such as Page 5 Line 170, Page 6 lines 201, 204, 225, and 238, Page 8 line 292.

Page 3 line 99 add comma before the word "but"

Page 6 line 233 add a period after the word "chemoradiotherapy" to avoid run-on sentence.

Page 8 line 313 add a period after the word "arena" and capitalize the word "However" to avoid run-on sentence.

Page 8 line 304 references hypoxia-activated prodrugs.  Consider adding a figure/table with a list of example prodrugs for the reader's reference without having to read reference article 128.

Page 2 line 92 The mTOR pathway is referenced.  Perhaps a brief explanation or description of this pathway and how hypoxia and lactate inhibits this would be helpful to the reader since this is a review article primarily being read by imagers and not cell biologists.

Conclusion: Excellent

References: Appropriate

Author Response

  1. Page 3 line 200 change "diving" cells to "dividing" cells

 “Diving” has been corrected to “dividing” which is now reflected on Line 109.

  1. Please use convention of punctuation in which comma is used before the word "and" in a list of words or independent phrases such as Page 5 Line 170, Page 6 lines 201, 204, 225, and 238, Page 8 line 292.

The convention of using a comma before the word “and” has been adopted in the areas noted throughout the manuscript and are now reflected on lines 193, 226, 228, 251, 265, and 387.

  1. Page 3 line 99 add comma before the word "but"

A comma has been added before the word “but” and can be found on line 108. 

  1. Page 6 line 233 add a period after the word "chemoradiotherapy" to avoid run-on sentence.

A period has been added after the word “chemoradiotherapy” and can be found on line 260.

  1. Page 8 line 313 add a period after the word "arena" and capitalize the word "However" to avoid run-on sentence.

A period has been added after the word “arena” and “However” has been capitalized which is now reflected on line 486.

  1. Page 8 line 304 references hypoxia-activated prodrugs.  Consider adding a figure/table with a list of example prodrugs for the reader's reference without having to read reference article 128.

Figure 1 has been added which features the discussed hypoxia activated prodrugs (HAPs) along with their chemical structures.

7. Page 2 line 92 The mTOR pathway is referenced.  Perhaps a brief explanation or description of this pathway and how hypoxia and lactate inhibits this would be helpful to the reader since this is a review article primarily being read by imagers and not cell biologists. 

A sentence was added to elaborate on the relevance of the mTOR pathway (Line 101). A more thorough explanation was avoided in order to mediate between other reviewer comments to shorten this section. 

Reviewer 4 Report

The authors provided a detailed summary of the functional imaging of hypoxia. However, there are a few questions and issues that I believed should be addressed before publication.   

1. Please verify the accuracy of the authors’ affiliations and email information.  

2. Please ensure the consistency in the style and font throughout the manuscript (e.g., lines 10-12, 93-94, and 330). 

3.  The authors should correct the term “XH4” to “HX4”.

      4. On page 5, lines 159-161, please revise these sentence as not all the listed radiopharmaceuticals are used for PET imaging.

Author Response

1. Please verify the accuracy of the authors’ affiliations and email information.  

Author affiliations and email information has been updated and verified.

2. Please ensure the consistency in the style and font throughout the manuscript (e.g., lines 10-12, 93-94, and 330). 

The differing font has been corrected to reflect the style throughout the rest of the manuscript. 

3. The authors should correct the term “XH4” to “HX4”.

“XH4” has been corrected to “HX4” throughout the manuscript. 

4. On page 5, lines 159-161, please revise these sentence as not all the listed radiopharmaceuticals are used for PET imaging.

The sentence has been revised to include only those pharmaceuticals used in PET imaging (Line 182).

Thank you again for your comments and constructive feedback.

Round 2

Reviewer 2 Report

p.10 l.381 ff. : The key explanantion on how photoacoustics can be used to detect tissue hypoxia is incorrect. Low oxygen levels can be detected using photoacoustic imaging without any external contrast agent. By using multiple wavelengths total hemoglobin concentration and oxygenation can be derived non-invasively and label-free based on the different optical absorption spectra of deoxygenated and oxygenated hemoglobin. The relative concentrations of these chromophores can then be calculated by spectroscopic inversion, giving estimates on total hemoglobin content and blood oxygenation.   p 7 l.241-242: "DCE-MRI employs the administration of a Gadolinium-based contrast agent after its distribution within the tissue." This sentence is incorrect. The contrast agents for DCE-MRI do not necssarily have to be gadolinium-based (it is just the most common one). A basic review on this topic can be found here: doi: 10.18632/oncotarget.16482.   p.11 l.421 ff: One of the key limitations of oxygen needle probes is missing, which is that the measurements are only point-based and do not provide any spatial information on tumor oxygenation.    

In general, there are also quite a few typos in the document (e.g. "environemnt" l.391, p.10, "tumor" instead of "tumors" p.11 l.408, ". ."  p.9 l.325 etc)

Author Response

Reviewer 2:

  1. p.10 l.381 ff. : The key explanation on how photoacoustics can be used to detect tissue hypoxia is incorrect. Low oxygen levels can be detected using photoacoustic imaging without any external contrast agent. By using multiple wavelengths total hemoglobin concentration and oxygenation can be derived non-invasively and label-free based on the different optical absorption spectra of deoxygenated and oxygenated hemoglobin. The relative concentrations of these chromophores can then be calculated by spectroscopic inversion, giving estimates on total hemoglobin content and blood oxygenation.

The key explanation and working principles of photoacoustic imaging has been revised and is now reflected beginning on page 11, line 383.

  1. p 7 l.241-242: "DCE-MRI employs the administration of a Gadolinium-based contrast agent after its distribution within the tissue." This sentence is incorrect. The contrast agents for DCE-MRI do not necssarily have to be gadolinium-based (it is just the most common one). A basic review on this topic can be found here: doi: 10.18632/oncotarget.16482.

This sentence has been revised to more accurately represent the role of gadolinium-based contrast agents in DCE-MRI and is now reflected on page 8, line 244.

  1. p.11 l.421 ff: One of the key limitations of oxygen needle probes is missing, which is that the measurements are only point-based and do not provide any spatial information on tumor oxygenation.

The limitation has been added to this section and is now reflected on page 12, line 429.

  1. In general, there are also quite a few typos in the document (e.g. "environemnt" l.391, p.10, "tumor" instead of "tumors" p.11 l.408, ". ." p.9 l.325 etc)

The document has been revised and the typos have been corrected using the track changes function which will be reflected on:

  • Page 11, line 400
  • Page 12, line 416
  • Page 9, line 329

Thank you again for your comments and constructive feedback.